# VC-Bench: Pioneering the Video Connecting Benchmark with a Dataset and Evaluation Metrics

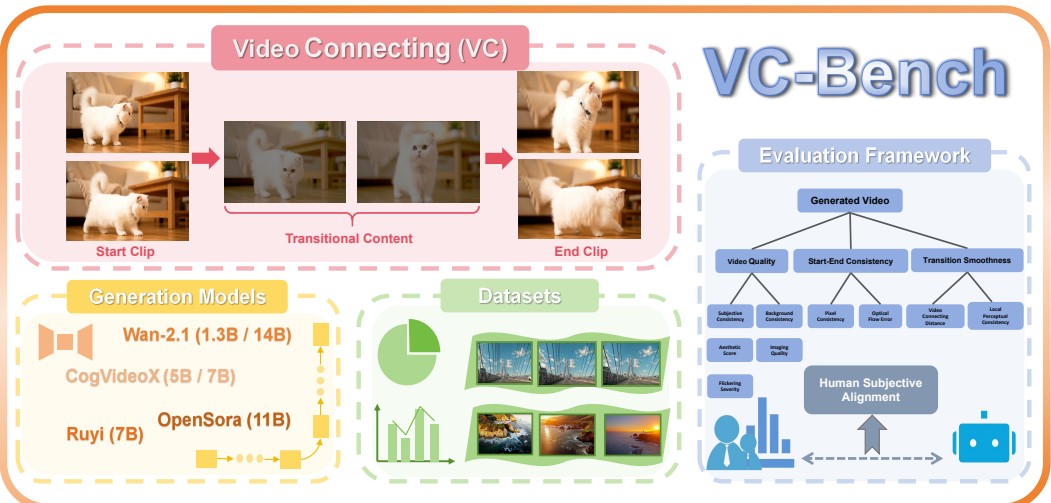

Figure 1: **VC-Bench Overview.** We introduce **VC-Bench**, a tailored benchmark for the novel **Video Connecting** task. We provide a precise definition of this task and adapt several open-source video generation models to support it. To establish a systematic and comprehensive evaluation framework, we curate a high-quality dataset spanning 15 categories. We develop 9 automated metrics to evaluate model performance across three critical dimensions: **Video Quality**, **Start-End Consistency**, and **Transition Smoothness**. Human subjective validation further confirms the framework's alignment with human preferences.

## ABSTRACT

While current video generation focuses on text or image conditions, practical applications like video editing and vlogging often need to seamlessly connect separate clips. In our work, we introduce **Video Connecting**, an innovative task that aims to generate smooth intermediate video content between given start and end clips. However, the absence of standardized evaluation benchmarks has hindered the development of this task. To bridge this gap, we proposed **VC-Bench**, a novel benchmark specifically designed for video connecting. It includes 1,579 high-quality videos collected from public platforms, covering 15 main categories and 72 subcategories to ensure diversity and structure. VC-Bench focuses on three core aspects: **Video Quality Score** $VQS$, **Start-End Consistency Score** $SECS$, and **Transition Smoothness Score** $TSS$. Together, they form a comprehensive framework that moves beyond conventional quality-only metrics. We evaluated multiple state-of-the-art video generation models on VC-Bench. Experimental results reveal significant limitations in maintaining start-end consistency and transition smoothness, leading to lower overall coherence and fluidity. We expect that VC-Bench will serve as a pioneering benchmark to inspire and guide future research in video connecting. The evaluation metrics and dataset are publicly available at: https://anonymous.4open.science/r/VC-Bench-1B67/.

# 1 INTRODUCTION

Recent advances in generative AI have transformed video creation, with models such as Sora (OpenAI, 2024), Runway-Gen (Runway, 2024), and Pika (PikaLabs, 2024) demonstrating unprecedented progress in text-to-video and image-to-video generation (Li et al., 2024; Sun et al., 2024b). These systems greatly enhance creative efficiency, enabling filmmakers and creators to produce concept videos within hours. Yet, existing efforts largely focus on generating standalone clips, leaving continuity-aware generation underexplored.

We introduce the task of **Video Connecting (VC)**: given two clips (a start and an end), the goal is to synthesize transitional content that is both spatio-temporally coherent and consistent with the originals. Unlike general video generation, VC requires strict semantic and visual alignment across discontinuous segments. This formulation captures practical needs—linking discrete shots in vlogs, filling surveillance gaps caused by device failures, or producing narrative transitions in film. More application of VC tasks are listed in Appendix C.2. By framing VC as a distinct problem, we highlight its potential to bridge the gap between isolated generation and real-world continuity demands.

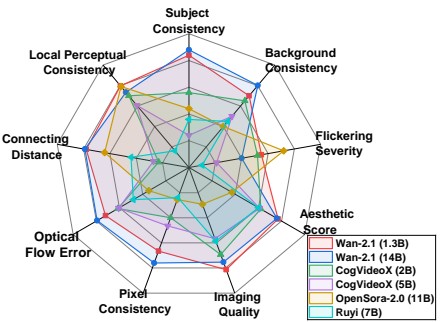

Figure 2: **Video Score on VC-Bench.** Based on VC-Bench, we evaluate the performance of 6 open source models on the Video Connection task. The experimental results can be seen in Table 2

Importantly, VC is not about arbitrarily linking any two unrelated clips. The task is meaningful when clips share a logical, semantic, or narrative relation, where continuity is expected but missing. To enable systematic study, we construct a benchmark covering diverse scenarios and propose multi-dimensional evaluation metrics along three axes: overall video quality, start–end consistency, and transition smoothness. We hope this work lays the foundation for advancing continuity-aware video generation.

The main contributions are three-fold:

- We are the first to systematically propose the novel task of **Video Connecting (VC)**, which expands the boundaries of video generation and lays a theoretical foundation for this emerging research direction.

- We construct a comprehensive benchmark for video connection, termed **VC-Bench**, which includes: a test video dataset covering multiple categories and diverse scenarios, as well as a three-dimensional video evaluation paradigm.

- Based on this benchmark, we conduct a comprehensive evaluation of multiple video generation models, perform an in-depth analysis of their performance, identify current technical limitations, and propose constructive research suggestions for future development.

# 2 RELATED WORKS

## 2.1 VIDEO GENERATION

**Text-to-Video Generation.** Text-to-video (T2V) generation is a technology that leverages AI models to transform natural language descriptions into video content. By understanding the semantics, scenes, and actions in text, it generates high-quality video sequences. In recent years, advancements in diffusion models and transformer architectures have significantly improved the fidelity and consistency of generated videos. Sora (OpenAI, 2024) can produce high-definition videos, excelling in complex scenes and multi-character interactions, while Kling AI (Kuaishou, 2024) offer motion control and style presets, suitable for creative generation. Vidu (ShengShu-AI, 2024) and Haiper (DeepMind, 2024) focus on narrative coherence and story generation capabilities. Wan2.1 (Wang et al., 2025) and HunyuanVideo (Kong et al., 2024) further advance efficient and cinematic video production, meeting diverse creative demands.

**Image-to-Video Generation.** Image-to-video(I2V) generation is capable of converting static images into dynamic videos. It is applicable to animation production and storytelling, often combined with text prompts to enhance flexibility. Runway Gen-3 Alpha (Runway, 2024) and Luma AI (LumaLabs, 2024) support I2V generation with high fidelity, while Jimeng AI (Jianying, 2024) focuses on visual effects for Chinese scenes, generating short videos. Ruyi (Team, 2024) and CogVideoX (Yang et al., 2024) support video generation based on the first and last frames. LTX-Video (HaCohen et al., 2024) can generate smooth 10s short videos in real time. Open-Sora 2.0 (Peng et al., 2025) integrate image and text inputs to produce high-quality dynamic content, significantly enriching visual storytelling possibilities.

## 2.2 Evaluations on Video Generation

In recent years, the rapid development of video generation technology has spawned a variety of metrics to evaluate the generated videos from different dimensions. The relevant evaluation metrics can be organized into the following dimensions:

**Movement dynamics evaluation:** For movement dynamics, multiple benchmarks (Ling et al., 2025) analyze motion naturalness through long-duration video datasets. MiraBench (Ju et al., 2024) builds a high-quality dataset of long-duration, fine-grained captions to better evaluate temporal consistency and motion intensity in video generation. (Liao et al., 2024) proposed DEVIL method, which evaluates the generation ability of the T2V model by defining fine-grained temporal dynamic scores and text prompts at multiple dynamic levels.

**Text alignment evaluation:** The generation model needs to strictly follow the semantic constraints of the text prompts. StoryBench (Bugliarello et al., 2023) systematically evaluates the ability to transfer from basic instruction understanding to complex story construction by setting up three levels of progressive tasks. StoryEval (Wang et al., 2024b) is specifically used to evaluate the model's ability to handle event-level story presentation. It uses VLMs to perform event-level decomposition and verification of generated videos, and integrates voting mechanisms to improve reliability.

**Time coherence evaluation:** Temporal coherence is a core challenge of video generation. ChronoMagic-Bench (Yuan et al., 2024) evaluates the rationality of the gradual process of the T2V model when generating high-dynamic time-lapse videos by introducing MTScore and CHScore. T2VBench (Ji et al., 2024) quantifies the performance of generated videos in terms of motion rationality and event timing logic with temporal dynamic features. TC-Bench (Feng et al., 2024) defines the initial and target state of the scene, and measures the transition integrity through the TCR metrics.

**Comprehensive performance evaluation:** The comprehensive evaluation system of video generation models is developing in a multi-dimensional and fine-grained direction. (Liu et al., 2024). AIGCBench(Fan et al., 2023) established the first scalable evaluation system for I2V generation tasks through 11 metricss. VBench(Huang et al., 2024) extends video evaluation from surface authenticity to intrinsic authenticity. T2V-CompBench(Sun et al., 2024a) integrating LLaVA-1.5 and object tracking technology, and showed significant advantages in complex prompt evaluation.

# 3 Video Connecting

## 3.1 Task Definition

Suppose the start video clip is denoted by $V_S = \{I_t\}_1^{N_S}$ and the end video clip is denoted by $V_E = \{I_t\}_1^{N_E}$, where $N_*$ denotes the frame counts of the video, and $I_t \in \mathbb{R}^{H \times W \times C}$ represents each frame. $V_S$ and $V_E$ share the same spatial resolution and frame rate. Given a pair of video clips $V_S$ and $V_E$ as well as an optional textual prompt $T$, the task of video connecting aims to generate a video $V$ satisfying to the following conditions:

- The start and end clips of $V$ are consistent with $V_S$ and $V_E$, respectively.
- $V$ holds the properties of the video, such as consistent spatial resolution and frame rate, motion smoothness, etc.

The first condition ensures that the generated video remains faithful to the provided start and end clips, denoted as $V_S$ and $V_E$. This constraint is critical in practical applications, as these clips are

| Benchmark Name | Year | Open-Domain | Text-Video | Image-Video | Video Connecting | Metrics |
|---|---|---|---|---|---|---|
| LFDM Eval (Ni et al., 2023) | 2023 | ✗ | ✔ | ✗ | ✗ | 3 |
| StoryBench (Bugliarello et al., 2023) | 2023 | ✔ | ✔ | ✗ | ✗ | 10 |
| CATER-GEN (Hu et al., 2024) | 2023 | ✗ | ✔ | ✔ | ✗ | 7 |
| SVD-Eval (Blattmann et al., 2023) | 2023 | ✔ | ✔ | ✔ | ✗ | 5 |
| EvalCrafter (Liu et al., 2024) | 2023 | ✔ | ✔ | ✔ | ✗ | 17 |
| VideoPhy (Bansal et al., 2024) | 2024 | ✔ | ✔ | ✗ | ✗ | 2 |
| PhyGenBench (Meng et al., 2024) | 2024 | ✔ | ✔ | ✗ | ✗ | 5 |
| MiraBench (Ju et al., 2024) | 2024 | ✔ | ✔ | ✗ | ✗ | 17 |
| DEVIL (Liao et al., 2024) | 2024 | ✔ | ✔ | ✗ | ✗ | 10 |
| StoryEval (Wang et al., 2024b) | 2024 | ✔ | ✔ | ✗ | ✗ | 7 |
| ChronoMagic-Bench (Yuan et al., 2024) | 2024 | ✔ | ✔ | ✗ | ✗ | 6 |
| T2VBench (Ji et al., 2024) | 2024 | ✔ | ✔ | ✗ | ✗ | 16 |
| TC-Bench (Feng et al., 2024) | 2024 | ✔ | ✔ | ✔ | ✗ | 7 |
| AIGCBench (Fan et al., 2023) | 2024 | ✔ | ✔ | ✔ | ✗ | 11 |
| Video-Bench (Han et al., 2025) | 2024 | ✔ | ✔ | ✔ | ✗ | 10 |
| Step-Video-T2V-Eval (Huang et al., 2025) | 2024 | ✔ | ✔ | ✗ | ✗ | 4 |
| VMBench (Lin et al., 2024) | 2025 | ✔ | ✔ | ✗ | ✗ | 5 |
| VBench 1.0 / 2.0 (Huang et al., 2024) | 2025 | ✔ | ✔ | ✔ | ✗ | 21 |
| T2V-CompBench (Sun et al., 2024a) | 2025 | ✔ | ✔ | ✗ | ✗ | 7 |
| **VC-Bench(Ours)** | **2025** | ✔ | ✔ | ✔ | ✔ | **9** |

Table 1: **The summary of benchmark in video generation.** We summarize the benchmark work related to video generation in the past three years and extract information.

typically predefined by content creators. The second condition is introduced to preserve the high visual quality of the output video throughout the generation process.

**Relation to FLF2V**  First-Last Frame to Video (FLF2V) (Wang et al., 2025; Zhang & Agrawala, 2025) generation is similar to our Video Connection (VC) task, as both control video content using start and end states. However, FLF2V is an Image-to-Video task, while VC is Video-to-Video, leading to distinct challenges. Details of the difference between these two tasks are listed in Appendix B.1.

**(1) Information Complexity:** In the FLF2V task, video generation is primarily guided by the information contained in the first and last frames, which provide static visual cues. In contrast, the VC task relies on the spatiotemporal information from the start and end video clips, which include dynamic motion, temporal context, and richer content. While this provides more information, it also introduces the challenge of effectively extracting and utilizing this complex spatiotemporal data, as processing video clips is significantly more demanding than handling static frames.

**(2) Content Difference:** In FLF2V task, the two frames typically belong to the same video sequence, exhibiting high consistency in content, style, and scene. The model only needs to focus on generating intermediate frames to ensure smooth visual and motion transitions. In contrast, for our video connecting task, the starting and ending videos may come from different video sequences, with potentially significant differences in content, such as lighting conditions, color schemes, and so on.

**(3) Temporal Consistency:** Both tasks require ensuring the temporal consistency. However, the challenge in video connecting lies in the fact that the start and end clips may have different motion patterns, rhythms. For instance, the starting video might have slow motion, while the ending video features fast motion. The model must generate a transitional video that smoothly adapts to the ending video's rhythm, which is more complex than simple frame interpolation.

## 3.2 Transfer Approach

To address the lack of support for video connection tasks in existing models, we adapted the Diffusion Transformer (DiT) architecture to generate videos conditioned on starting and ending segments, enhancing temporal consistency, semantic coherence, and content controllability. The methods are:

**(1) Mapping to Latent Space:** The start and end clips are mapped to the latent space's initial and final positions. The intermediate section, filled with random noise, is processed by the DiT model alongside conditioned segments, guiding the denoising process to model temporal dependencies and ensure spatiotemporal consistency.

**(2) SLERP for Conditional Control:** Using spherical linear interpolation (SLERP), as in TVG (Zhang et al., 2024), we blend features of the start and end clips to generate smooth transition parts, improving cross-scene video connections for natural transitions.

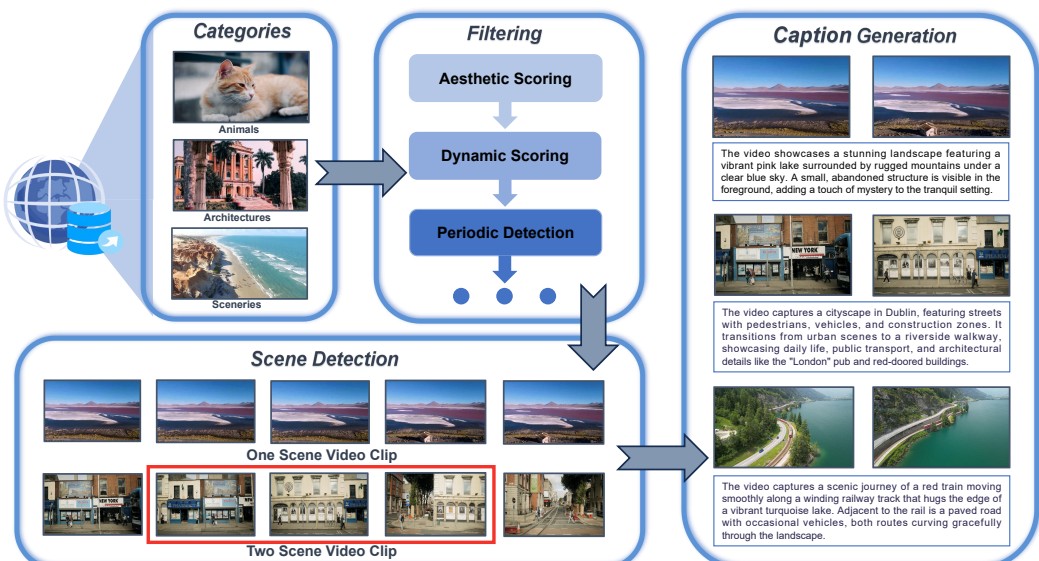

Figure 3: **Dataset Construction.** We outline the construction proces s of the our dataset: **Web Data Crawling and Classification**: Organized into 15 major categories and 72 subcategories. **Data Filtering**: Selected high-quality test data through aesthetic scoring, dynamic scoring, and periodic detection. **Scene Detection**: Employed PySceneDetect for scene segmentation. **Caption Generation**: Utilized Qwen2-VL to generate high-quality caption.

**Remarks**   The DiT architecture for video generation has limitations in achieving hard-constrained generation: **(1) VAE Encoder Scale Effect:** Fixed-ratio compression and techniques like rounding or zero-padding in VAE encoders cause irreversible loss of original content. **(2) Patch-Based Serialization:** Dividing video into fixed-size spatiotemporal patches with positional encoding alters the original video's characteristics, especially at boundary patches. **(3) Latent Space Denoising:** Compressing video into a low-dimensional latent space for efficient denoising, then reconstructing it, introduces noise and reduces fidelity.

## 4   VC-BENCH

In this section, we will introduce VC-Bench in detail, including the construction of the dataset and the establishment of evaluation metrics. The benchmarks related to video generation are summarized in Table 1. Compared with previous work that mainly focuses on the evaluation of T2V and I2V tasks, our benchmark focuses on comprehensive evaluation of video generation models.

### 4.1   DATASET CONSTRUCTION

The VC-Bench dataset is meticulously curated from multiple public video resource platforms such as Pexels, Pixabay, Mixkit, and YouTube. The construction pipeline is illustrated in Figure 3.

In addition to conventional steps including aesthetic filtering, dynamic motion screening, watermark removal, and caption generation, we introduce additional procedures tailored to the video connecting task: Scene Detection, Periodic Motion Detection, and Video Clips Extraction.

**Scene Detection**: Automated analysis of raw videos is performed using the PySceneDetect scene detector to ensure each video contains either a single scene or exactly two scenes. This is based on the rationale that video connectivity between any number of scenes can be achieved by connecting videos with two scenes.

**Periodic Motion Detection**: Videos with prominent periodic motions (e.g., spring oscillations, dumbbell lifts) are excluded, as such content does not adequately reflect the nature of the video connectivity task. Structural Similarity Index (SSIM) is used to compute the similarity between the first frame and each subsequent frame. Peaks are detected via SciPy(Virtanen et al., 2020), and the period is estimated by calculating the average frame difference between consecutive peaks.

**Video Clips Extraction**: We set the total video duration to 5 seconds, with the start and end clips varying between 2 to 4 seconds. For videos containing two scenes, we ensure that transition frames are excluded from these clips. This is designed to evaluate the capability of existing video generation models in reconstructing the transitional portions of the videos.

We present detailed statistics of the dataset to provide a comprehensive understanding. Our dataset consists of a total of 1,579 videos. In terms of video duration, the videos length ranges from 4 seconds to 43 seconds. Notably, videos lasting between 7.5 and 20 seconds are the most prevalent. Regarding video quality, all videos have a resolution exceeding 720p and were filtered using the aesthetic score predictor, achieving an average aesthetic score as high as 0.55, as depicted in Figure 4. Additionally, we employed Qwen2-VL (Bai et al., 2023; Wang et al., 2024a) to equip each video with detailed and expressive textual descriptions. The length of these captions ranges from 18 to 57 characters.

For video categorization, we employed a BERT-based text classification model to classify videos using their textual descriptions, significantly reducing dataset construction time. As shown in Figure 4, the dataset spans 15 major categories and 72 subcategories, ensuring diverse content.

## 4.2 Evaluation Metrics

Given the characteristics of video connection tasks, it is necessary to design specialized metrics to evaluate the quality of video content. It should be noted that due to the limitations of current conditional video generation models, achieving seamless transitions between the starting and ending video clips remains challenging, rendering traditional evaluation methods insufficient. Therefore, we propose our **VC-Bench** to more accurately assess video generated content, providing a more reliable basis for research and optimization in video generation technology.

Our work proposes a set of comprehensive metrics for evaluating the quality of video generation, covering the following three aspects: **Video Quality Score** $VQS$, **Start-End Consistency Score** $SECS$ and **Transition Smoothness Score** $TSS$. The details are as follows.

**Video Quality Score** $VQS$**:** Inspired by VBench (Huang et al., 2024), we propose a comprehensive video quality evaluation method based on five key metrics to assess the realism and overall performance of generated videos.

(1) *Subject Consistency* $Q_S$: This metric evaluates whether the main subject (e.g., person, animal, or object) in the video maintains consistent identity, appearance, and structure throughout the sequence.

(2) *Background Consistency* $Q_B$: It is used to measures the stability of the background over time, ensuring no unnatural shifts or jitters.

(3) *Flickering Severity* $Q_F$: Quantifying flickering artifacts in local details (e.g., color or brightness), indicating instability in generation.

(4) *Aesthetic Score* $Q_A$: Automatically predicting the artistic quality and visual appeal of the video using a pre-trained aesthetic assessment model.

(5) *Imaging Quality* $Q_I$: Assessing low-level visual artifacts in individual frames (e.g., blur, distortion, or noise).

The numerical range of the above metrics scores is between 0 and 1, then $VQS$ is calculated as the average of these mstrics.

$$VQS = [Q_S + Q_B + (1 - Q_F) + Q_A + Q_I]/5. \tag{1}$$

**Start-End Consistency Score** $SECS$**:** The VC task imposes higher demands on maintaining consistency between the start and end video clips. This dimension primarily evaluates the consistency of the generated video at the pixel level between its start and end clips.It comprises two metrics:

(1) *Pixel Consistency* $C_P$: Evaluating the pixel alignment between the start or end clips of the generated video and the original video by comparing luminance, contrast, and structural similarity.

(2) *Optical Flow Error* $C_{OF}$: Optical flow describes the motion information of pixels between frames. This metric measures the motion consistency between the generated and the original video.

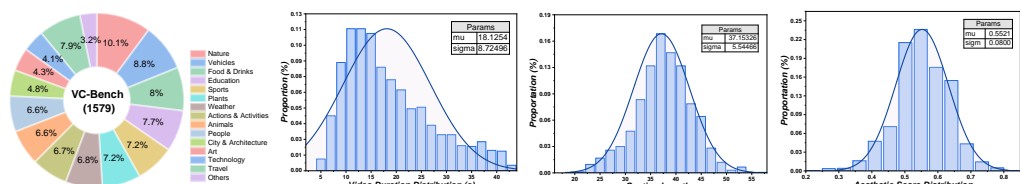

Figure 4: **Distribution of Curated Data.** We analyzed the dataset, including category statistics, video duration statistics (24fps), caption length statistics, and aesthetic score statistics. Statistical analysis demonstrates that our dataset exhibits high quality, diversity, and complexity, meeting the rigorous requirements for benchmark testing.

Similarly, $SECS$ is calculated as follows.

$$SECS = [C_P + (1 - C_{OF})]/2. \tag{2}$$

**Transition Smoothness score** $TSS$**:** Evaluating whether the transition between the original clips and generated part is natural and fluid, consisting of two metrics:

(1) *Video Connecting Distance* $T_{CD}$*:* Aligning the generated and the original video frame in time with DTW method. The transition consistency is measured by calculating the ratio of the structural similarity $SSIM(f_I, f_G)$ between the frame $f_I$ in the original video and the frame $f_G$ in the generated video and the frame spacing $d$. It ensures that the transition part can naturally connect the start and end video clips while avoiding abrupt changes.

(2) *Local Perceptual Consistency* $T_{LP}$*:* Quantifying the perceptual similarity between adjacent frames in the transition region by extracting features using a pre-trained VGG network and computing the LPIPS distance. This determines whether perceptual jumps exist at the transition point, thereby assessing the natural fluidity of the generated video's transition.

$TSS$ is formulated below.

$$TSS = [(1 - T_{CD}) + T_{LP}]/2. \tag{3}$$

Finally, the comprehensive evaluation score of the generated video is expressed as the mean of the above three scores. We provide the specific calculation formula for each metrics in the Appendix B.2.

$$Score = (VQS + SECS + TSS)/3. \tag{4}$$

## 5 EXPERIMENTS

In this section, we evaluate some state-of-the-art video generation models using VC-Bench. We begin by introducing the evaluation setup, followed by presenting quantitative comparison results. We also assess the quality of connected videos under varying durations of the start and end clips. Finally, we analyze the alignment between our proposed evaluation metrics and subjective human judgments.

### 5.1 SETTINGS

We begin with a brief description of the experimental setup used in our study; more detailed configurations are provided in the supplementary materials.

**Models.** We evaluated six mainstream video generation models on the video connectivity task: Wan2.1 (Wang et al., 2025) (1.3B and 14B), CogVideoX (Yang et al., 2024) (2B and 5B), Open-Sora 2.0 (Peng et al., 2025) (11B), and Ruyi (Team, 2024) (7B).

**Framework.** The DiT architecture and a 3D causal variational autoencoder were adopted and modified according to the method described in Section 3.2. The framework was adapted to the video connectivity task through latent space mapping of the start and end video clips and interpolation-based conditional control.

**Conditioning setting.** We selected the following conditioning settings: Using single frame, 1 second (24 frames), 1.5 seconds (36 frames), and 2 seconds (48 frames) as the start and end clip. The text prompt was set as optional. In all cases, a 5-second transition video was generated.

| | Video Quality | | | | |
|---|---|---|---|---|---|
| | Subject Consistency | Background Consistency | Flickering Severity (↓) | Aesthetic Score | Imaging Quality |
| Wan-2.1(1.3B) | 0.921 | 0.944 | 0.045 | 0.577 | 0.720 |
| Wan-2.1(14B) | 0.922 | 0.946 | 0.060 | 0.576 | 0.713 |
| CogVideoX(2B) | 0.914 | 0.943 | 0.048 | 0.562 | 0.704 |
| CogVideoX(5B) | 0.906 | 0.940 | 0.079 | 0.561 | 0.685 |
| Open-Sora 2.0(11B) | 0.911 | 0.938 | 0.028 | 0.537 | 0.644 |
| Ruyi(7B) | 0.909 | 0.939 | 0.090 | 0.560 | 0.688 |

| | Start-End Consistency | | Transition Smoothness | | Total Score |
|---|---|---|---|---|---|
| | Pixel Consistency | Optical Flow Error (↓) | Connecting Distance (↓) | Local Perceptual Consistency | |
| Wan-2.1 (1.3B) | 0.933 | 0.042 | 0.022 | 0.839 | 0.892 |
| Wan-2.1 (14B) | 0.952 | 0.031 | 0.021 | 0.820 | 0.893 |
| CogVideoX (2B) | 0.880 | 0.058 | 0.077 | 0.810 | 0.864 |
| CogVideoX (5B) | 0.893 | 0.059 | 0.073 | 0.782 | 0.858 |
| OpenSora-2.0 (11B) | 0.851 | 0.098 | 0.036 | 0.837 | 0.859 |
| Ruyi (7B) | 0.848 | 0.078 | 0.056 | 0.649 | 0.827 |

Table 2: **Video Score on VC-Bench.** Based on VC-Bench, we evaluate the performance of 6 open source models on the Video Connection task.

| | Same Start & End Scenes | | | | | Distinct Start & End Scenes | | | | |
|---|---|---|---|---|---|---|---|---|---|---|
| Conditional proportion | FLF2V | 40% | 60% | 80% | Avg. | FLF2V | 40% | 60% | 80% | Avg. |
| Wan-2.1 (1.3B) | 0.840 | 0.870 | 0.872 | 0.874 | 0.864 | 0.801 | 0.832 | 0.836 | 0.835 | 0.826 |
| Wan-2.1 (14B) | 0.851 | 0.871 | 0.873 | 0.875 | 0.868 | 0.802 | 0.822 | 0.831 | 0.834 | 0.822 |
| CogVideoX (2B) | 0.820 | 0.838 | 0.845 | 0.847 | 0.838 | 0.760 | 0.786 | 0.797 | 0.797 | 0.785 |
| CogVideoX (5B) | 0.785 | 0.834 | 0.847 | 0.849 | 0.829 | 0.734 | 0.786 | 0.799 | 0.802 | 0.780 |
| OpenSora-2.0 (11B) | 0.756 | — | 0.844 | 0.840 | 0.813 | 0.765 | — | 0.763 | 0.765 | 0.764 |
| Ruyi (7B) | 0.827 | 0.824 | 0.816 | 0.820 | 0.822 | 0.767 | 0.771 | 0.771 | 0.766 | 0.769 |

Table 3: **Comparison of Duration Ratio.** We conducted comparative experiments on 6 test models to evaluate the impact of start-end duration ratios on generation performance. The experimental settings included FLF2V, 40% (24 frames each), 60% (36 frames each), and 80% (48 frames each), with the generated video length fixed at 5 seconds (120 frames).

## 5.2 MAIN RESULTS

We evaluated six video generation models adapted for the Video Connecting task using our VC-Bench. To standardize evaluations, we converted negative metrics to positive by subtracting from 1 and normalized metrics with ranges to the 0–1 interval. Results are presented in Table 2 and Figure 2.

As shown in Table 2, Wan-2.1 (1.3B and 14B) outperforms other models, achieving the highest total score. It excels in subject consistency, background consistency, optical flow error, and connection distance, demonstrating superior spatiotemporal coherence and transition naturalness. Conversely, CogVideoX (2B and 5B) performs strongly in aesthetic score and imaging quality but is less effective in flickering severity and transition smoothness, which is suitable for short videos prioritizing visual appeal. Open-Sora-2.0 (11B) leads in flickering severity, effectively suppressing instabilities like color or brightness fluctuations due to its fine-tuned noise control, but it shows moderate performance in start-end consistency. Ruyi (7B) performs poorly in transition smoothness due to inadequate training for the Video Connecting task, resulting in abrupt transitions that reduce overall fluency.

In summary, current video generation models demonstrate strong discriminative capabilities on our VC-Bench. However, they still underperform significantly in terms of start-end consistency and transition smoothness. These shortcomings highlight the ongoing challenges in maintaining temporal coherence and ensuring fluid transitions between video clips in the Video Connecting task. Moving forward, future research will prioritize conditional video generation, exploring advanced techniques and optimized training strategies to enhance overall performance and effectiveness in this domain.

## 5.3 ANALYSIS

**Comparison between One Scene and Two Scenes.** As shown in Table 3, we evaluated the quality of transition videos generated by the model under two conditions: when the start and end scenes are the same, and when they differ. The results indicate that the quality of transitions generated in the former scenario is consistently higher than in the latter. This suggests that two-scene video connecting remains highly challenging for current video generation models. The models struggle to "imagine" plausible transitions between disparate start and end clips, often producing significant "hallucinations" in the generated videos—such as characters from the end clip appearing abruptly without logical context—which clearly deviate from realistic and coherent motion continuity.

| Score | Objective Avg. | Subjective Avg. | Correlation Coefficient | Subjective Consistency |
|-------|----------------|-----------------|-------------------------|------------------------|
| VQS  | 0.815 | 0.807 | 0.839 | 0.968 |
| SECS | 0.911 | 0.916 | 0.905 | 0.980 |
| TSS  | 0.867 | 0.824 | 0.812 | 0.979 |

Table 4: **Validate VC-Bench's Human Alignment.** 30 volunteers were invited to subjectively score video generation results on Video Quality, Start-End Consistency, and Transition Smoothness. We then calculated the correlation coefficient between objective and subjective scores to demonstrate that objective scores effectively reflect human preferences. Additionally, we computed the ICC metric among volunteers to confirm high scoring consistency.

**Impact of Conditional Proportion.** As shown in Table 3, we conducted experiments to analyze how the quality of connected videos changes as the proportion of conditioning frames from the start and end clips increases. Overall, video quality improves gradually with a higher ratio of conditioning frames, with Wan-2.1 (14B) consistently outperforming other models. It is noteworthy that the quality of videos generated with higher conditioning frame ratios (40%, 60%, and 80%) is significantly better than that achieved using only the start and end frames. This improvement can be attributed to the richer spatiotemporal features and motion consistency captured by the model when provided with more contextual frames.

**Case Presentation and Analysis.** We have carefully selected cases from multiple categories for presentation and analysis, with details provided in the Appendix D.3. These cases vividly illustrate the common issues in Video Connecting, including subject distortion (involving Animations, Animals, and Humans), artifact flickering (such as Plants and Sports), image distortion (can be seen in citeps), and unnatural motion trajectories (like Sports). This indicates that the adaptability of existing models is still insufficient, necessitating further research and technological advancements in the future.

**Human Alignment of VC-Bench.** To validate our VC-Bench test set's alignment with human subjective perceptions, we engaged 30 volunteers to manually evaluate generated videos. In Table 4, correlation coefficients showed strong similarity between objective and subjective evaluations, confirming that VC-Bench effectively reflects human perceptions and can serve as an automated video evaluation. To ensure subjective consistency and detect potential malicious scoring, we applied the ICC(2,K) statistical method (Koo & Li, 2016). Results confirmed high rater reliability, with no evidence of intentional high or low scores.

## 6 CONCLUSION AND FUTURE WORK

In this paper, we introduce Video Connecting, a novel task in conditional video generation, and propose VC-Bench, a benchmark tailored to evaluate performance in this domain. Unlike existing benchmarks focusing on general video quality, VC-Bench assesses Video Quality, Start-End Consistency, and Transition Smoothness using a diverse dataset of 1,579 high-quality samples. Evaluations of open-source models reveal challenges in spatiotemporal coherence and transition fluency, especially in cross-scenes settings. Limitations include testing only 5-second videos, leaving longer sequences unexplored, and reliance on open-source models due to the absence of closed-source support. Future work should include longer videos and closed-source models for robust validation. We aim for VC-Bench to drive advancements in Video Connecting generative models.

## 7 ETHICS STATEMENT

This work complies with the ICLR Code of Ethics. The proposed VC-Bench are constructed entirely from publicly available videos that are free of personally identifiable or sensitive information. All data sources follow their original licenses, and the processing steps are documented in Section 4.1. Our benchmark and evaluation framework are designed to foster research on controllable and coherent video generation, with positive applications in areas such as film production, short video creation, advertising, and virtual reality. We acknowledge the general risk that generative video models could be misused to produce misleading or harmful content; however, our contribution is limited to a benchmark and evaluation metrics, which are intended solely for advancing academic research. All procedures were conducted under the approval of an institutional ethics review board, and all participants provided informed consent prior to participation. The data collected from human subjects

were limited to annotation feedback and subjective quality ratings of videos, without collection of any personally identifiable or sensitive information. No additional ethical risks beyond those typically associated with generative modeling research have been identified.

# 8 REPRODUCIBILITY STATEMENT

We have taken several steps to ensure the reproducibility of our work. The construction and pre-processing steps of the VC-Bench dataset are described in Section 4.1. The detailed description of the proposed method and evaluation metrics can be found in Sections 4.2 of the main paper and further clarified in Appendix B.1. Moreover, we provide an anonymous link to our source code and evaluation scripts at https://anonymous.4open.science/r/VC-Bench-1B67/ to facilitate reproduction of the reported results.

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

## A  THE USE OF LARGE LANGUAGE MODELS (LLMS)

In accordance with the policy on the use of large language models (LLMs), we disclose that LLMs were utilized as an assistive tool during the preparation of this manuscript. The LLM (specifically, DeepSeek) was employed exclusively for the purpose of enhancing the clarity, fluency, and overall readability of the narrative text.

The precise role of the LLM was limited to:

- Text Polishing and Refinement: Rewording and rephrasing sentences to improve grammatical correctness and stylistic flow.
- Improving Coherence: Assisting in ensuring smooth transitions between paragraphs and sections.

It is critical to emphasize that the LLM did not contribute to the core intellectual content of this work. All research ideation, methodological design, data analysis, interpretation of results, scientific conclusions, and the original drafting of the manuscript were conducted solely by the human authors.

| Video Generation Tasks | Conditions | Input Length | Motion Info | Scene | Information Complexity |
|---|---|---|---|---|---|
| T2V | Text prompt | Text only | Yes (implicit) | Free | High |
| I2V | Initial image | 1 frame | No | Same | Medium |
| Video Extension | Initial video segment | ∼24-48 frames | Yes | Same | Medium |
| Frame Interpolation | Two adjacent frames | 2 frames | No | Same | Small |
| FLF2V | First/Last frames | 2 frames | No | Same | Medium |
| VC | Two video segments | ∼48-96 frames | Yes | Distinct | High |

Table 5: Difference between VC Task and Other Video Generation Tasks.

# B  METHODOLOGY AND VC TASK CLARIFICATION

## B.1  DIFFERENCE BETWEEN VC TASK AND OTHER VIDEO GENERATION TASKS

Compared to other video generation tasks such as T2V, I2V, video extension, and frame interpolation, the proposed VC task is considerably more demanding. For instance, T2V relies solely on a text prompt with implicitly inferred motion and allows the model to generate a free-scene video, while I2V conditions on a single frame without motion information, both maintaining relatively lower information requirements. Video extension takes a short continuous video segment (24–48 frames) from the same scene and focuses on extending the temporal span, whereas frame interpolation and FLF2V operate on only two adjacent or boundary frames, requiring no explicit motion cues and remaining within a fixed scene, resulting in small to medium complexity.

In contrast, VC takes two full video segments of approximately 48–96 frames each as input, both containing rich temporal motion patterns and originating from distinct scenes. This configuration obligates the model to not only understand and preserve the motion dynamics within each segment, but also to bridge semantic gaps, reconcile scene differences, and generate coherent transitions. Such requirements elevate VC into a highly complex and information-intensive task, making it substantially more challenging than previous video generation settings. Difference between VC task and other video generation tasks are shown in Tabel 5

## B.2  DETAILS IN EVALUATION METRICS

In this section, we will introduce the various metrics in our benchmark and their implementation methods in detail.

(1) *Subject Consistency $Q_S$:* This metric assesses whether the primary subject (e.g., a person, animal, or object) in a video maintains consistent identity, appearance, and structure throughout the sequence. It ensures that, across the entire video, the same subject remains coherent. In the generated video, the subject present in the input's start and end frames should be preserved, while also maintaining consistency during the intermediate transition part. We specifically adopt the Dino model to implement subject consistency detection. Because Dino model has not been trained and is unable to classify subjects and ignore their intra-class differences, it is particularly sensitive to changes in subject differences in the video and is suitable to detect subject changes. The calculation formula is as follows:

$$Q_S = \frac{1}{N-2} \sum_{t=2}^{N-1} \frac{1}{3} \left( D \left\langle I_1, I_t \right\rangle + D \left\langle I_{t-1}, I_t \right\rangle + D \left\langle I_t, I_N \right\rangle \right), \tag{5}$$

where $N$ is the number of video frames, $I_t$ is the $t-th$ frame, and $D \left\langle \cdot, \cdot \right\rangle$ is the dot product of the Dino feature cosine similarity between the two frames. In general, we detect whether the current frame maintains the subject consistency by calculating the average cosine similarity of each frame with the first frame, the previous frame, and the last frame of the video. Finally, we average all the detected frames as our final subject consistency score.

(2) *Background Consistency $Q_B$:* In addition to subject consistency, the video background also needs to be consistent. This metrics is used to measure the stability of the background over time to ensure that there are no unnatural shifts or jitters in the video. Specifically, we calculate the CLIP feature

map for each frame of the video, and then similar to the subject consistency, we calculate the average CLIP feature cosine similarity of each frame with the first frame, previous frame, and last frame of the video to detect the consistency of the background. Finally, we average all the detected frames as our final background consistency score. The formulation is as follows:

$$Q_B = \frac{1}{N-2} \sum_{t=2}^{N-1} \frac{1}{3} \left( C \langle I_1, I_t \rangle + C \langle I_{t-1}, I_t \rangle + C \langle I_t, I_N \rangle \right), \tag{6}$$

where $C \langle \cdot, \cdot \rangle$ is the dot product of the CLIP feature cosine value between the two frames.

(3) *Flickering Severity $Q_F$:* In the generated videos, there are often rapid and irregular changes in brightness, color, texture, or content between consecutive frames, resulting in a sense of visual instability. This phenomenon is particularly common in video generation and can seriously affect the viewing experience. Common flickering includes brightness flickering and color flickering. In our metrics, we first divide the image into non-overlapping areas, and then determine whether there is flickering between adjacent frames from the image's YUV space (focusing on image brightness changes) and HSV space (focusing on image color changes). We calculate the proportion of the flickering area between two frames using the following formula:

$$r_t = \frac{\sum_{P_t \in S_t} I((\mathcal{L} \langle P_t, P_{t+1} \rangle + \mathcal{C} \langle P_t, P_{t+1} \rangle)/2 > \eta)}{|S_t|}. \tag{7}$$

Among them, $S_t$ represents the set of non-overlapping regions that the $t-th$ frame is divided into, and $P_t$ is the divided region. $I(\cdot)$ is an indicator function. If the input is greater than the threshold $\eta$, it takes 1, otherwise it takes 0. It is mainly used to determine whether it is a flickering area based on the degree of change in a certain area. $\mathcal{L} \langle \cdot, \cdot \rangle$ represents the brightness change of the same area in two adjacent frames in the YUV space, and $\mathcal{C} \langle \cdot, \cdot \rangle$ represents the color change in the HSV space. If the mean of the two change values is greater than the threshold, the area is considered to be flickering. Through the ratio definition, the proportion of the flickering area is obtained. Finally, we can calculate the average flicker ratio $\bar{r}_t$ of the video to indicate the flickering serverity of the video. It should be noted that this metrics is a negative indicator and needs to be positively processed during visualization.

(4) *Aesthetic Score $Q_A$:* It is used to quantify the visual beauty of images or videos are widely used in image generation, video synthesis, photography evaluation and other fields. It measures the aesthetic quality of content, such as composition, color, and thematic appeal, through algorithms or manual scoring. We used the pretrained image aesthetic quality predictor LAION to score each frame aesthetically (0-10), calculated the average aesthetic score, and finally linearly normalized the score to 0-1.

(5) *Imaging Quality $Q_I$:* It mainly considers the frame-level generation quality in the generated video, focusing on issues such as distortion, blur, and noise in the frame image. We use the pretrained MUSIQ model as an image quality predictor to score each frame. Finally, the average score of the entire video sequence is calculated and divide by 100 to normalize to the range of $0-1$.

(6) *Pixel Consistency $C_P$:* This metrics is used to evaluate the pixel-level alignment ability of the start and end clips of the generated video. Specifically, for each pair of original and generated frames $(I_t, \hat{I}_t)$ at corresponding positions, we use SSIM as metrics to judge the pixel-level similarity between the two frames. SSIM comprehensively considers pixel-level features such as brightness, contrast, and structural similarity, and can well evaluate the pixel consistency maintained by the starting and ending videos. Finally, we take the average value as an metrics of video pixel consistency. The calculation formula is as follows:

$$C_P = \frac{1}{N_s + N_e} \sum_{t=1}^{N_s+N_e} SSIM(I_t, \hat{I}_t), \tag{8}$$

where $N_s$ and $N_e$ represent the number of frames of the starting and ending videos.

(7) *Optical Flow Error $C_{OF}$:* Optical flow describes the per-pixel motion between consecutive video frames, representing how each pixel moves in a 2D plane over time. This metric is crucial for evaluating motion consistency between a generated video and its original counterpart, ensuring that

dynamic elements (e.g., object movement, camera motion) align realistically. Specifically, for each pair of original and generated frames $(I_t, \hat{I}_t)$ at corresponding positions, we calculate their optical flow $OF()$ and calculate the average optical flow error. It is worth noting that the value range of the optical flow error is not in the range of $0 - 1$, but since our optical flow window size is 16, it is easy to know that the maximum optical flow error is 32 (based on Manhattan Distance), we divide it by 32 to normalize it to $0 - 1$.

$$C_{OF} = \frac{1}{N_s + N_e - 1} \sum_{i=1}^{N_s + N_e - 1} \left| OF(I_t) - OF(\hat{I}_t) \right|, \qquad (9)$$

(8) *Video Connecting Distance* $T_{CD}$: This metrics is used to measure the connection distance between the middle generated part and the first and last videos. Specifically, we first utilize the DTW method to align the frames of the generated video with the start and end clips of the original video. Then, for a pair of corresponding frames $(I, I_G)$ of the starting (ending) video, we randomly select the frame $I_M$ of the middle generated part and calculate $SSIM \langle I, I_M \rangle$ and $SSIM \langle I_G, I_M \rangle$ respectively. We hope that the difference between these two values is small. However, considering that the frame distances from $I_M$ to $I$ and $I_G$ may be different, we use $SSIM/d$ as the measurement standard, where $d$ is the frame distance between the two frames. The formula is as follows:

$$T_{CD} = \left| \frac{SSIM \langle I, I_M \rangle}{d_{I_M \to I}} - \frac{SSIM \langle I_G, I_M \rangle}{d_{I_M \to I_G}} \right|. \qquad (10)$$

We select $K$ groups of corresponding frames from the starting and ending videos and $Z$ frames from the middle generated part for calculation respectively, and finally calculate the average value as the representation of this metrics.

(9) *Local Perceptual Consistency* $T_{LP}$: To rigorously evaluate the naturalness of transitions in generated videos, we propose a perceptual similarity metric based on deep feature extraction and LPIPS. This method focuses on detecting abrupt perceptual discontinuities between adjacent frames in transition regions (e.g., scene cuts or interpolated clips), which are critical for assessing temporal coherence. Specifically, we first use the VGG model to extract features frame by frame, then calculate the absolute difference of the features between every two frames, and take the average as the perceptual error of the two frames. Finally, we average all frames to get the perceptual error of the video. Since this error is a negative metrics and its value is between 0 and 1, we use 1 minus this value as our local perceptual consistency metrics. The perceptual error formula is as follows:

$$\varepsilon = Mean(|\mathcal{V}(I_t) - \mathcal{V}(I_{t-1})|). \qquad (11)$$

Among them, $\mathcal{V}(\cdot)$ represents the spatiotemporal features extracted using VGG model, and $Mean()$represents the mean value according to the feature dimension.

### B.3 Construction for the Subjective Human Evaluation

We invited 30 volunteers and asked them to focus on three aspects such as Video Quality, Start-End Consistency, and Transition Smoothness for assessment. A 10-point scale was used to rate the video quality (1 being "extremely poor" and 10 being "perfect"). The human subjective score is calculated through subjective evaluation method. The Instruction text and Screenshots for the subjective human evaluation are shown in Figure 5

## C Expansion of Experiments

### C.1 Extension to Multi-Clip Video Connecting

VC-Bench is designed naturally supports more complex scenarios, including multi-clip connections and looped video generation. To illustrate this, we evaluate Wan2.1 on a tiny multi-clip test set containing 2-, 3-, 4-, and 5-segment connections. Experimental results are shown in Table 6

### C.2 Application of Video Connecting

The Video Connecting task has wide applicability across creative and industrial contexts as shown in Figure 6. In filmmaking and visual effects, VC can generate transitional shots or smooth scene

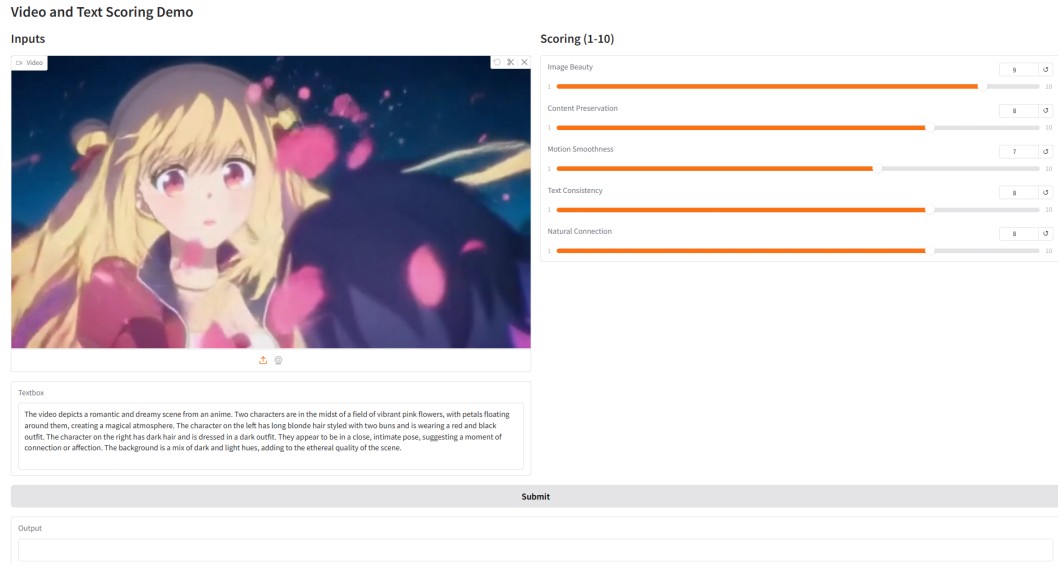

Figure 5: Instruction text and screenshots for the subjective human evaluation.

| Metrics | Two Clips | Three Clips | Four Clips | Five Clips |
|---|---|---|---|---|
| Subject Consistency | 0.844 | 0.836 | 0.828 | 0.835 |
| Background Consistency | 0.915 | 0.909 | 0.913 | 0.914 |
| Flickering Severity (↓) | 0.278 | 0.262 | 0.290 | 0.255 |
| Aesthetic Score | 0.538 | 0.542 | 0.554 | 0.545 |
| Imaging Quality | 0.625 | 0.630 | 0.623 | 0.629 |
| Pixel Consistency | 0.914 | 0.908 | 0.917 | 0.916 |
| Optical Flow Error (↓) | 0.095 | 0.083 | 0.102 | 0.073 |
| Connecting Distance (↓) | 0.143 | 0.149 | 0.155 | 0.142 |
| Local Perceptual Consistency | 0.515 | 0.524 | 0.518 | 0.516 |

Table 6: Extension to Multi-clip Connecting.

changes, reducing the need for costly reshoots and post-production. For short-form video platforms such as TikTok or YouTube Shorts, VC enables creators to link fragmented clips into coherent narratives, enhancing fluency and viewer engagement. Advertising campaigns can also benefit from seamless transitions between product-focused and storytelling segments. Beyond traditional media, VC supports immersive experiences in virtual and augmented reality, where continuity is critical for maintaining realism and user immersion. Even in personal media such as travel vlogs, VC allows amateur creators to connect discontinuous scenic shots into smooth and engaging stories. These applications underscore VC as a versatile tool for continuity-aware video generation.

## D  ANALYSIS AND ROBUSTNESS OF VC-BENCH

### D.1  SENSITIVITY ANALYSIS

Since all open-source models generate videos with a fixed duration of 5 seconds, extending video length would naturally degrade scores and is therefore predictable. We hope to objectively evaluate the optimal performance of the community models. Instead, we analyze sensitivity to video resolution using three output resolutions: 640×320, 960×432, and 1024×576, evaluated on the same models. Experimental results are shown in Table 7.

Among all metrics, only **Aesthetic Score** shows statistically significant sensitivity to video resolution (F=6.18, p=0.020). All other metrics exhibit no significant differences (p > 0.17), demonstrating strong robustness of our evaluation framework.

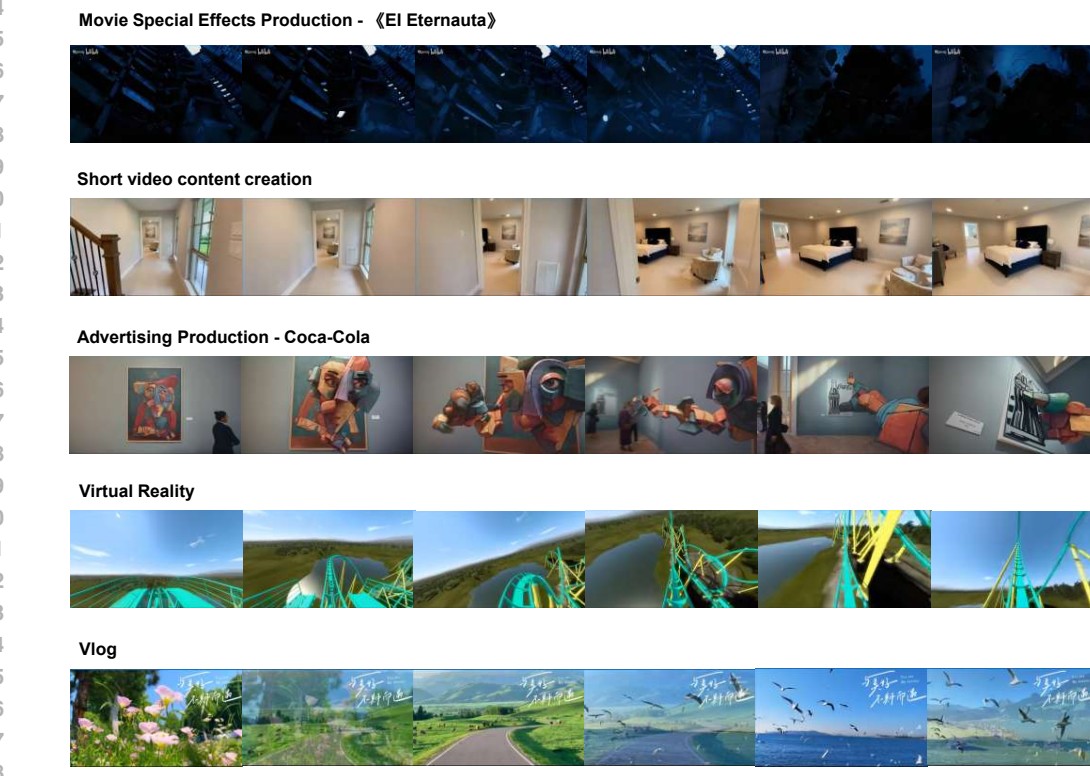

**Movie Special Effects Production - 《El Eternauta》**

**Short video content creation**

**Advertising Production - Coca-Cola**

**Virtual Reality**

**Vlog**

Figure 6: Application of Video Connecting

| Metric | F-value | p-value |
|---|---|---|
| Subject Consistency | 0.146 | 0.866 |
| Background Consistency | 0.777 | 0.488 |
| Flickering Severity (↓) | 0.081 | 0.923 |
| **Aesthetic Score** | **6.179** | **0.020** |
| Imaging Quality | 2.126 | 0.175 |
| Pixel Consistency | 0.005 | 0.995 |
| Optical Flow Error (↓) | 1.235 | 0.336 |
| Connecting Distance (↓) | 0.169 | 0.847 |
| Local Perceptual Consistency | 0.005 | 0.995 |

Table 7: Sensitivity Analysis of Metrics to Resolution.

## D.2 METRICS INTERPRETABILITY REPRESENTATION

In this section, we select specific cases to demonstrate the interpretability of some metrics, including consistency metrics (Subject Consistency/Background Consistency) and Transition Smoothness Metrics (Connection Distance). See Table 8, Figure 7 and Figure 8 for specific examples , it can be seen that the metrics we provide can effectively reflect human subjective evaluations.

## D.3 CASE STUDY

We present generated samples from different models in the video transition task, covering various categories, with each category containing positive and negative examples. The issues in the generated results are annotated in the case figures, including inconsistent characters, object deformation, temporal discontinuity, flickering, and so on. The details are showed in Figure 9

| Case Type | VCD Value | Qualitative Example | Phenomenon Description |
|-----------|-----------|---------------------|------------------------|
| Better | 0.011 | A small lion following behind a hippo in water | The video is coherent and stable, with smooth object motion and no flickering. |
| Medium | 0.025 | The scene focuses on a person on a beach | The switch between the front and back views of the person is abrupt, but the main structure remains stabel. |
| Poor | 0.052 | A cartoon panda in front of a street shop | Temporally incoherent, with obvious object deformation and content jumping in transition frames. |

Table 8: Qualitative Examples of Video Connecting Distance.

**Video Connecting Distance (↓)**

**Good Generation** with VCD=0.011

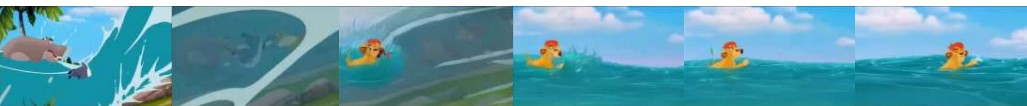

In a vibrant animated scene, a rhino swims through clear blue water, followed by a playful lion with red hair. The lion enjoys the ocean, set against a bright sky with fluffy clouds.

**Medium Generation** with VCD=0.025

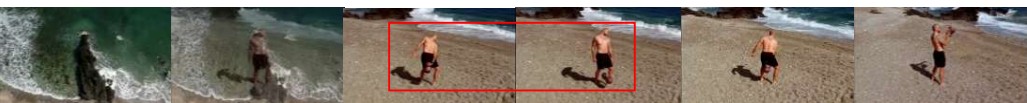

The video begins with an aerial view of waves crashing against rocks in the ocean. It then transitions to a beach where a man, shirtless and wearing black shorts, is seen performing exercises with a medicine ball, showcasing strength and agility.

**Bad Generation** with VCD=0.052

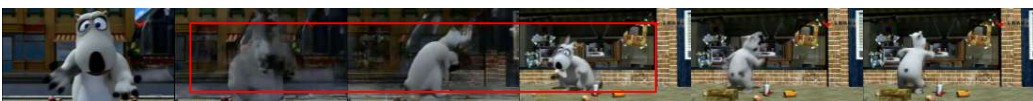

In a chaotic urban setting, a cartoon raccoon character panics and runs through debris. The background features a truck and damaged storefronts with broken glass and scattered cans. The scene is filled with urgency and disorder, highlighting the raccoon's frantic escape amidst destruction.

Figure 7: Qualitative Examples of Video Connecting Distance.

**Subject and Background Consistency**

**Good Generation**

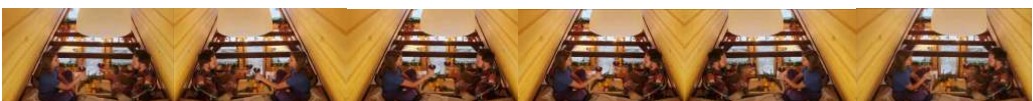

A couple sits inside a cozy, wooden A-frame cabin, toasting with wine glasses. They are surrounded by warm lighting and festive decorations, creating an intimate and romantic atmosphere. (Wan-2.1 1.3B, Subject Consistency **0.982**, Background Consistency **0.969**)

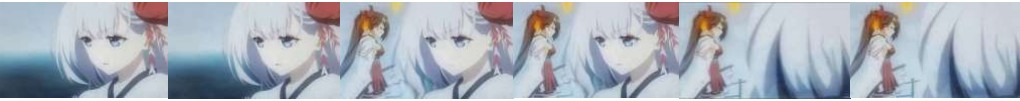

The video captures a serene scene of white daisies swaying gently in the breeze. The soft focus highlights the delicate petals and vibrant yellow centers, set against a blurred green background, creating a peaceful and natural atmosphere. (Wan-2.1 1.3B, Subject Consistency **0.941**, Background Consistency **0.959**)

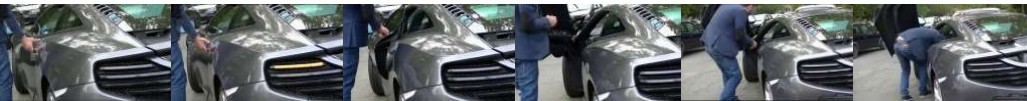

The video captures a serene coastal landscape at dusk, featuring a narrow strip of land with fields and a winding road bordered by the calm sea. The sky is painted with soft hues of pink and orange, transitioning to a darker blue as it meets the horizon. The scene exudes tranquility and natural beauty.
(Wan-2.1 14B, Subject Consistency **0.958**, Background Consistency **0.918**)

**Subject and Background Consistency**

**Terrible Generation**

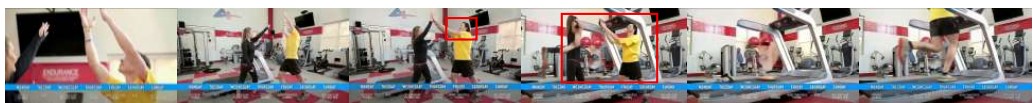

In a gym, a man and woman high-five while walking on treadmills. The video shows a workout schedule with days labeled from Monday to Sunday, including strength training, rest, and running sessions. (Wan-2.1 1.3B, Subject Consistency **0.731**, Background Consistency **0.863**)

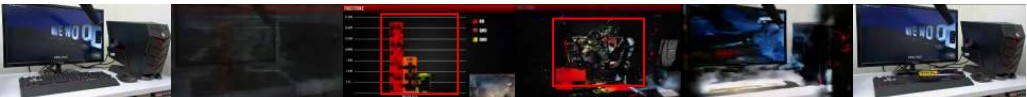

The video showcases a gaming setup with an MSI GTX 1070 graphics card, displaying ""Hollywood"" in a game. It includes performance metrics and a bar graph comparing HD, QHD, and UHD resolutions, highlighting the card's capabilities. (CogVideoX 2B, Subject Consistency **0.761**, Background Consistency **0.815**)

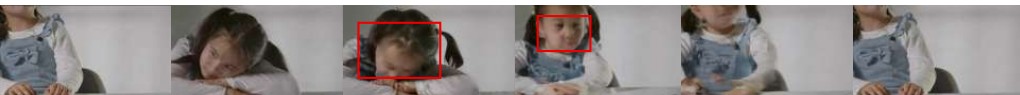

A young girl with pigtails, wearing a blue dress and white shirt, sits at a table. She leans forward, resting her head on her arms, appearing thoughtful or tired. Her expression is contemplative as she looks down at the table. (CogVideoX 5B, Subject Consistency **0.837**, Background Consistency **0.855**)

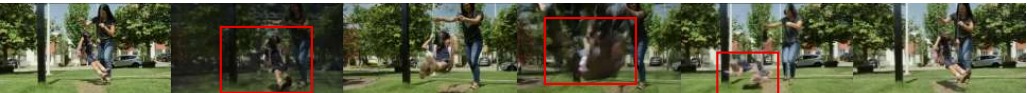

A woman pushes a young girl on a swing in a park. The girl swings high, laughing and enjoying the ride. The woman smiles, ensuring the girl's safety as they move back and forth. The scene is set against a backdrop of green grass, trees, and parked cars, capturing a joyful moment of play.
(CogVideoX 5B, Subject Consistency **0.812**, Background Consistency **0.867**)

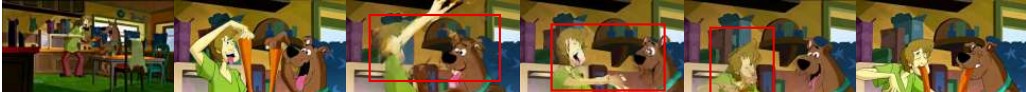

In a cartoon scene, two characters, one human and one dog, hold large carrots. The human stretches the carrot above their head, while the dog mimics the action. Both appear excited, with wide smiles and raised eyebrows, set against a colorful, indoor background. (Ruyi, Subject Consistency **0.728**, Background Consistency **0.905**)

Figure 8: Qualitative Examples of Consistency Metrics.

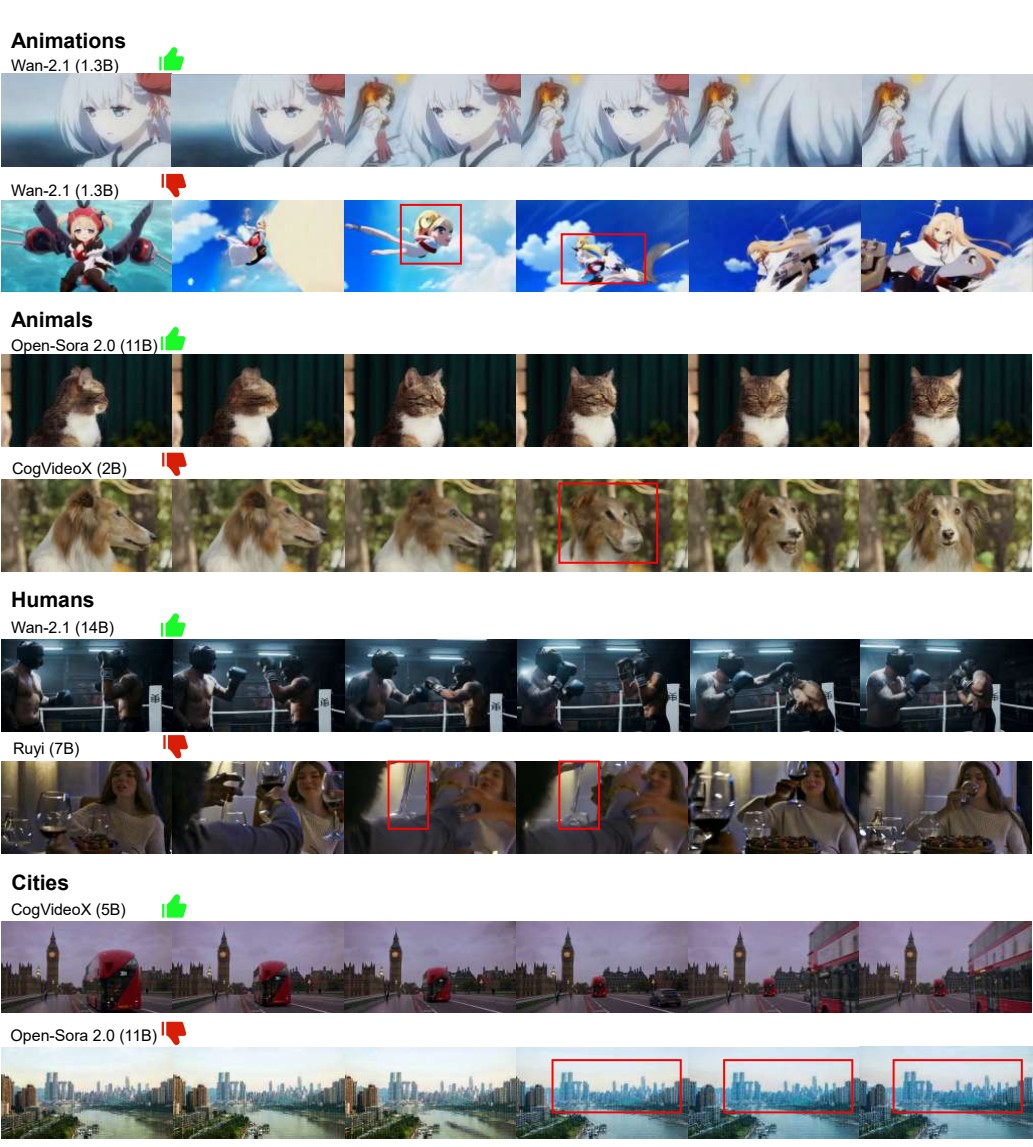

Figure 9: Case Study.

