# OpenReview forum: "VC-Bench: Pioneering the Video Connecting Benchmark with a Dataset and Evaluation Metrics"
_ICLR.cc/2026/Conference — Submitted to ICLR 2026_

### Official Review · Reviewer_u86t · 2025-10-31

**Soundness:** 2
**Presentation:** 3
**Contribution:** 2
**Rating:** 4
**Confidence:** 4

**Summary:**

The paper introduces the novel task of Video Connecting, which aims to generate transitional video content that seamlessly links a given start clip and end clip. The authors propose VC-Bench, a new benchmark to evaluate this task, consisting of a 1579 video dataset and a three part evaluation framework (Video Quality Score, Start-End Consistency Score, and Transition Smoothness Score). The paper provides a comprehensive evaluation of several video generation models on this benchmark, identifying current limitations in start-end consistency and transition smoothness.

**Strengths:**

- The paper formalizes the Video Connecting task, which, while related to existing video generation problems, presents a non-trivial challenge. The paper provides a valuable comparison by adapting and evaluating several recent state-of-the-art video generation models for this new task.

- The paper offers a detailed pipeline for the VC-Bench dataset construction and the calculation of the proposed evaluation metrics.

**Weaknesses:**

- The long-term impact of the benchmark may be limited, as Video Connecting could be viewed as a niche or minor task rather than a foundational problem. There is significant overlap with existing video generation, extension, or interpolation tasks, and few works are specifically dedicated to this problem, which may limit the benchmark's adoption.

- The dataset construction pipeline (e.g., scene detection, clip filtering, captioning) and core evaluation metrics (particularly the Video Quality Score components) are largely adopted from existing benchmarks for text-to-video and image-to-video generation. The newly proposed task-specific metrics, such as the Start-End Consistency Score and Transition Smoothness Score, appear to be straightforward implementations and lack significant novelty.

- The proposed SECS and TSS metrics rely on a direct comparison against the ground-truth video. For a generative task, there are potentially many plausible ways to connect two clips. Relying on ground-truth similarity may unfairly penalize novel or creative, thus making the metrics less reliable for evaluating the true generative capabilities of a model.

- Related to the point above, while the paper distinguishes the VC task from First-Last Frame to Video generation, the evaluation metrics do not seem to fully capture the complexity of ensuring content consistency with the entirety of the start and end clips, instead focusing on pixel-level and optical flow comparisons which are still largely frame-based.

**Questions:**

- Regarding the human alignment evaluation (Section 5.3): Did the authors just check the correlation with human scores, or did they actively try to make the metrics match human preferences? For instance, how were the weights for the sub-metrics (in VQS, SECS, TSS) decided? Were they tuned to match human scores, or just set by a simple rule, like averaging?

---

> ### Author Response · Authors · 2025-11-27
> **We appreciate the reviewers' suggestions. In response to your valuable comments, we have provided detailed explanations in our reply.**
>
> ```
> The long-term impact of the benchmark may be limited, as Video Connecting could be viewed as a niche or minor task rather than a foundational problem. There is significant overlap with existing video generation, extension, or interpolation tasks, and few works are specifically dedicated to this problem, which may limit the benchmark's adoption.
> ```
>
> To address this concern, we clarify that VC task represents a significant and distinct challenge rather than other tasks.
>
> **1. Practical application.**
> Video connecting is widely used in many real-world applications, such as short video generation, vlogs, advertising, film production, et. As reviewer Qn9h noted, this direction has a promising research potential.
>
> **2. Fundamental difference.**
>  As summarized in the below, VC differs significantly from other tasks in terms of input structure, motion information, scene consistency and information complexity. Thus, VC task constitutes a high-difficulty video generation task.
>
> **3. Bridging the research gap.**
> As you noted, few works specifically target video connecting. Our paper provides a formal task definition, a comprehensive evaluation system, and a reusable dataset construction pipeline.
>
> |                   | Conditions          | Input Length      | Motion Info | Scene        | Information Complexity |
> | ----------------- | ------------------- | ----------------- | ----------- | ------------ | ---------------------- |
> | **Extension**     | Start video clip    | ~24–48 frames     | Yes         | Same         | Medium                 |
> | **Interpolation** | Two adjacent frames | 2 frames          | No          | Same         | Small                  |
> | **FLF2V**         | First/Last frames   | 2 frames          | No          | Same         | Medium                 |
> | **VC**            | **Two video clips** | **~48–96 frames** | **Yes**     | **Distinct** | **High**               |
>
> ```
> The dataset construction pipeline (e.g., scene detection, clip filtering, captioni  ng) and core evaluation metrics (particularly the Video Quality Score components) are largely adopted from existing benchmarks for text-to-video and image-to-video generation. The newly proposed task-specific metrics, such as the Start-End Consistency Score and Transition Smoothness Score, appear to be straightforward implementations and lack significant novelty.
> ```
>
> There appears to be a misunderstanding regarding the novelty of our proposed VC-Bench. To clarify, our core contributions lie in the following three aspects.
>
> **1.  Reproducible dataset pipeline.**
> Our proposed dataset construction pipeline is fully reproducible and provides a practical toolchain that future researchers can reuse directly.
>
> **2. Targeted Dataset.**
> Unlike typical video datasets based on the same scene, VC-Bench intentionally includes cross-scene and cross-semantic clip pairs, better reflecting real workflows.
>
> **3. Task-specific metric design.**
>
> - **VQS** draws upon VBench, as video generation tasks share commonalities in this dimension, making the adoption of established metrics both reasonable and necessary.
> - **SECS** and **TSS** metrics are specifically designed for our VC task. They respectively address maintaining consistency between the start and end video clips and ensuring smooth transitions between them.

---

> > ### Author Response · Authors · 2025-11-27
> >
> > ```
> > The proposed SECS and TSS metrics rely on a direct comparison against the ground-truth video. For a generative task, there are potentially many plausible ways to connect two clips. Relying on ground-truth similarity may unfairly penalize novel or creative, thus making the metrics less reliable for evaluating the true generative capabilities of a model.
> > ```
> >
> > We appreciate your comments. We understand the importance of novelty and plausibility in video generation. To address this, we clarify as follows:
> >
> > - **TSS** evaluates the temporal consistency of the transition video itself, not its similarity to ground-truth video. As long as the transition is smooth and coherent, it can achieve a reasonable score, thereby avoiding constraints on the generative model's creativity.
> > - **SECS** aims to evaluate the consistency of the start and end video clips. Thus, comparing with the ground truth is both reasonable and complaint with the task definition. SECS allows for creative video generation.
> >
> > ```
> > Related to the point above, while the paper distinguishes the VC task from First-Last Frame to Video generation, the evaluation metrics do not seem to fully capture the complexity of ensuring content consistency with the entirety of the start and end clips, instead focusing on pixel-level and optical flow comparisons which are still largely frame-based.
> > ```
> >
> > Our proposed benchmark systematically integrates assessment dimensions from VBench, specifically including subject consistency and background consistency. These metrics leverage VLMs like DINO and CLIP to extract high-level semantic features from generated outputs, enabling effective evaluation for global consistency. To demonstrate that, we supplement qualitative examples of consistency metrics, available in revised paper (Appendix D.2).
> >
> > ```
> > Regarding the human alignment evaluation (Section 5.3): Did the authors just check the correlation with human scores, or did they actively try to make the metrics match human preferences? For instance, how were the weights for the sub-metrics (in VQS, SECS, TSS) decided? Were they tuned to match human scores, or just set by a simple rule, like averaging?
> > ```
> >
> > To clarify, our goal was to design an unbiased, and universally applicable benchmark. The alignment with human judgment is a validation of our proposed design, not a result of tuning towards it.
> >
> > - **We did not perform any manual weight adjustment or training-based optimization** to avoid introducing subjective bias.
> > - As described in the main text (Formula (4)), we employ an **equal-weight averaging** approach to combine the sub-metrics: VQS, SECS, and TSS.
> > - In Section 5.3, we report **significant and consistent correlations** between our proposed metrics and human subjective scores in the table below The results demonstrate that these three sub-metrics effectively reflect human preferences.
> >
> > We conducted a human evaluation with 30 volunteers to validate VC-Bench.
> >
> > |      | Objective Avg. | Subjective Avg. | Correlation Coefficient |
> > | ---- | -------------- | --------------- | ----------------------- |
> > | VQS  | 0.815          | 0.807           | 0.839                   |
> > | SECS | 0.911          | 0.916           | 0.905                   |
> > | TSS  | 0.867          | 0.824           | 0.812                   |
> >
> > Experimental results demonstrate that our VC-Bench evaluation outcomes align with human subjective assessments.

---

### Official Review · Reviewer_WfZb · 2025-10-31

**Soundness:** 3
**Presentation:** 3
**Contribution:** 3
**Rating:** 6
**Confidence:** 4

**Summary:**

This paper introduces VC-Bench, a novel benchmark for evaluating models on the emerging Video Connecting (VC) task — generating smooth, temporally coherent transitions between given start and end video clips. The benchmark includes a curated dataset of 1,579 high-quality videos spanning 15 major and 72 subcategories, along with 9 quantitative metrics that evaluate three key dimensions: Video Quality, Start-End Consistency, and Transition Smoothness. Results highlight the open-source models’ limitations in maintaining continuity and temporal smoothness, while human evaluation shows strong alignment with the proposed metrics.

**Strengths:**

- Novel Task Definition: Clear formulation of the Video Connecting task as a distinct challenge, bridging isolated generation and temporal continuity
- Comprehensive Benchmark Design: A well-curated dataset with rigorous filtering, aesthetic scoring, and scene detection ensures quality and diversity.

**Weaknesses:**

- Model Diversity: Evaluation excludes closed-source systems (e.g., Sora, Runway Gen-3) that might exhibit different performance trends.
- Metric Interpretability: Some metrics (e.g., Video Connecting Distance) could benefit from additional qualitative examples to illustrate their perceptual meaning.
- Minor Writing Artifacts: Occasional typographical spacing and minor stylistic inconsistencies could be refined.

**Questions:**

- How sensitive are the evaluation metrics (especially TSS and SECS) to different video lengths and resolutions?
- Could VC-Bench be extended to evaluate multi-clip or looped video transitions beyond simple start-end pairs?

---

> ### Author Response · Authors · 2025-11-27
> **Thank you for your insightful comments and recognition of our work. The questions you raised are highly constructive, and we have conducted in-depth analysis and supplementary experiments based on your suggestions. Below is our response to your comments.**
>
> ```
> Model Diversity: Evaluation excludes closed-source systems (e.g., Sora, Runway Gen-3) that might exhibit different performance trends.
> ```
>
> Thanks for your comments. For the model diversity, the existing experiments showed the evaluations of six open-source models, which covered a diversity of open-source models; It is difficult for the closed-source models (e.g., Sora 2 and Luma Ray3) to support the proposed VC task because they just provide the interfaces of T2V, I2V and FLF2V generation.
>
> We approximatly evaluate the proposed VC task on the closed-source models, Sora 2 and Luma Ray3, by the closest video-generation setting conditioned on FLF2V. The first and last frames originate from the first and last clips of the proposed VC task. The following table shows results:
>
> |               | Subject Consistency | Background Consistency | Flickering Severity (↓) | Aesthetic Score | Imaging Quality | Local Perceptual Consistency |
> | ------------- | ------------------- | ---------------------- | ----------------------- | --------------- | --------------- | ---------------------------- |
> | **Sora 2**    | **0.878**           | **0.936**              | **0.051**               | **0.553**       | **0.680**       | **0.786**                    |
> | **Luma Ray3** | **0.942**           | **0.934**              | **0.157**               | **0.553**       | **0.680**       | **0.654**                    |
>
> |                              | Same Scene |               | Distinct Scenes |               |
> | ---------------------------- | ---------- | ------------- | --------------- | ------------- |
> | **Models**                   | **Sora 2** | **Luma Ray3** | **Sora 2**      | **Luma Ray3** |
> | Subject Consistency          | 0.922      | 0.948         | 0.834           | 0.935         |
> | Background Consistency       | 0.945      | 0.947         | 0.926           | 0.920         |
> | Flickering Severity (↓)      | 0.034      | 0.142         | 0.067           | 0.172         |
> | Aesthetic Score              | 0.582      | 0.570         | 0.523           | 0.531         |
> | Imaging Quality              | 0.693      | 0.678         | 0.667           | 0.690         |
> | Local Perceptual Consistency | 0.790      | 0.697         | 0.781           | 0.611         |
>
> The results showed that the performance trends of closed-source models align with open-source models to some extent.
>
> ```
> Metric Interpretability: Some metrics (e.g., Video Connecting Distance) could benefit from additional qualitative examples to illustrate their perceptual meaning.
> ```
>
> We appreciate this suggestion. In the revised version (Appendix D.2), we have added qualitative examples that may help illustrate the alignment of our metrics with human perception.
>
> ```
> How sensitive are the evaluation metrics (especially TSS and SECS) to different video lengths and resolutions?
> ```
>
> **Regrading video length**, current open-source models usually generate 5-second clips, their ability to model longer durations is limited; hence degrade performance in long video generation is predictable.
>
> **Regarding video resolution,** VC-Bench is resolution-insensitive. We test three resolutions (640×320, 960×432, 1024×576).P-value is used to evalate the sensitiveness of video resolution. As shown below:
>
> |                              | p-value   |
> | ---------------------------- | --------- |
> | Subject Consistency          | 0.865     |
> | Background Consistency       | 0.488     |
> | Flickering Severity (-)      | 0.922     |
> | **Aesthetic Score**          | **0.020** |
> | Imaging Quality              | 0.175     |
> | Pixel Consistency            | 0.994     |
> | Optical Flow Error (-)       | 0.335     |
> | Connecting Distance (-)      | 0.847     |
> | Local Perceptual Consistency | 0.994     |
>
> Among all metrics, only the aesthetic score showed sensitivity to resolution changes (p=0.020), consistent with the understanding that higher resolutions align better with human aesthetic preferences. All other metrics demonstrated robustness (p-values all >0.17), fully validating VC-Bench's exceptional robustness.
>
> ```
> Could VC-Bench be extended to evaluate multi-clip or looped video transitions beyond simple start-end pairs?
> ```
>
> Thank you for comments. VC-Bench can naturally extend to multi-clip condition.
>
> To more clearly evaluate model performance, we selected a subset of the VC-Bench dataset, partitioning samples into four conditions: 2-, 3-, 4-, and 5-clip. The experimental results are shown below:
>
> | Clip Number | Two   | Three | Tour  | Tive  |
> | ----------- | ----- | ----- | ----- | ----- |
> | VQS         | 0.729 | 0.731 | 0.726 | 0.734 |
> | SECS        | 0.909 | 0.912 | 0.908 | 0.921 |
> | TSS         | 0.686 | 0.688 | 0.682 | 0.687 |
>
> The experimental results demonstrate that our proposed VC-Bench exhibits strong adaptability for multi-clip video connecting tasks, effectively evaluating the quality of generated videos in terms of temporal logic and content coherence.

---

### Official Review · Reviewer_Qn9h · 2025-11-03

**Soundness:** 2
**Presentation:** 2
**Contribution:** 2
**Rating:** 2
**Confidence:** 4

**Summary:**

This paper proposes a benchmark for the video connecting task, which requires video generation models to synthesize the missing intermediate frames between the start and end video clips. Specifically, this paper apply scene detection, periodic motion detection, and video clips extraction to construct the datasets, which provides the diversity on open-domains. This paper further proposes 9 metics based on video quality, start-end consistency and transition smoothness score. Comprehensive experiments are conducted on various video genernation models.

**Strengths:**

1. The proposed task is highly valuable and meaningful for future research.
2. This paper is well-organized and easy to read.

**Weaknesses:**

1. The paper needs a more in-depth comparative analysis beyond the task definition to clarify what specific innovations have been made in constructing this benchmark, especially compared with the existing First-Last Frame to Video task.
2. Compared with the existing First-Last Frame to Video task, what different requirements does the video connecting task impose on the generation model? Are there corresponding experiment results to support this in the evaluation?

**Questions:**

Please see the weakness.

---

> ### Author Response · Authors · 2025-11-27
> **Thank you for your understanding of our proposed Video Connecting (VC) task and the construction of our VC-Bench. We have addressed each of your comments individually.**
>
> ```
> The paper needs a more in-depth comparative analysis beyond the task definition to clarify what specific innovations have been made in constructing this benchmark, especially compared with the existing First-Last Frame to Video task.
> ```
> Our contributions can be summarized into three aspects.
> 1. **A more challenging task.**
>    Unlike traditional video generation tasks, video connection involves greater information complexity, higher content variability, and more difficult-to-maintain temporal consistency. Since the start and end video clips inherently contain temporal information and may originate from different scenes, this task imposes higher demands compared to generating videos from First Last Frame to Video (FLF2V) task.
> 2. **Systematic benchmark for VC task.**
>    Unlike existing benchmarks, our proposed VC-Bench comprises three dimensions: VQS, SECS, and TSS. VQS serves as a universal metric for video evaluation. Additionally, our VC task requires maintaining consistency of the start and end video clips while ensuring smooth transitions. Hence, we designed two specialized dimensions: SECS and TSS.
> 3. **Comprehensive and in‐depth experimental analysis.**
>    We evaluated six open-source models on our proposed benchmark, conducting experiments from multiple aspects including proportion of start/end clips (Section 5.2 of the paper), scene dissimilarity between start and end clips (Section 5.2 of the paper), multi-clip generalization (Appendix C.1), and resolution sensitivity (Appendix D.1).
>
> ```
> Compared with the existing First-Last Frame to Video task, what different requirements does the video connecting task impose on the generation model? Are there corresponding experiment results to support this in the evaluation?
> ```
> Our proposed VC task imposes higher demands on generative models in terms of representation and rusion capabilities of spatio-temporal information and cross-scene semantic understanding compared to FLF2V task.
> 1. **Representation and Fusion Capabilities of Spatiotemporal Information**
> - FLF2V Task: Input consists of static images containing only spatial information, enabling the model to learn relatively simple motion trajectories from the start frame to the end frame.
> - VC Task: VC Task: The temporal and spatial information provided by the start and end clips reduces ambiguity in inferring intermediate states. This enables the model to perform feature fusion and video generation based on more comprehensive information.
> 2. **Cross-Scene Semantic Understanding Capability**
> - FLF2V Task: Inputs typically originate from the same scene. The model's primary task is to infer continuous motion of the subject under the same scene.
> - VC Task: Inputs may originate from different scenes, viewpoints, or even involve distinct subjects. The model must achieve a coherent transition by integrating scene understanding with semantic reasoning.
> Experimental Setup: To validate spatio-temporal information enhancement, we selected six open-source models and simulated increasing information levels by adjusting the proportion of start and end clips. These were set to 2 frames (FLF2V), 24 frames (40%), 36 frames (60%), and 48 frames (80%). FLF2V can be considered a special case. To compare cross-scene semantic understanding capabilities, we tested on same- and difstinct-scene. All generated videos were uniformly 5 seconds (120 frames) in length. The experimental results are as follows.
>
> |                    | Same Start & End Scenes |         |         |         | Distinct Start & End Scenes |         |         |         |
> | ------------------ | ----------------------- | ------- | ------- | ------- | --------------------------- | ------- | ------- | ------- |
> | Condition          | FLF2V                   | VC(40%) | VC(60%) | VC(80%) | FLF2V                       | VC(40%) | VC(60%) | VC(80%) |
> | Wan-2.1(1.3B)      | 0.840                   | 0.870   | 0.872   | 0.874   | 0.801                       | 0.832   | 0.836   | 0.835   |
> | Wan-2.1(14B)       | 0.851                   | 0.871   | 0.873   | 0.875   | 0.802                       | 0.822   | 0.831   | 0.834   |
> | CogVideoX(2B)      | 0.820                   | 0.838   | 0.845   | 0.847   | 0.760                       | 0.786   | 0.797   | 0.797   |
> | CogVideoX(5B)      | 0.785                   | 0.834   | 0.847   | 0.849   | 0.734                       | 0.786   | 0.799   | 0.802   |
> | Open-Sora 2.0(11B) | 0.756                   | /       | 0.844   | 0.840   | 0.765                       | /       | 0.763   | 0.765   |
> | Ruyi(7B)           | 0.827                   | 0.824   | 0.816   | 0.820   | 0.767                       | 0.771   | 0.771   | 0.766   |
>
> **Result:** Results validate that the enhanced spatio-temporal information can improve model performance. Besides, the generation outcomes for VC significantly outperform those for FLF2V. Moreover, the experiments reveal limitations in the model's cross-scene semantic understanding.

---

### Meta-Review · Area_Chair_7gq9 · 2025-12-22

**Summary:**

This paper introduces a new task called video connection and proposes corresponding benchmarks and evaluation metrics. In the first round, this paper received three reviews (642). The reviewers primarily raised questions about the distinction between this paper and existing video tasks, such as First-Last Frame, interpolation , and extension, which undermines its significance. The evaluation metrics also heavily rely on frame-based approaches, lacking representativeness. Therefore, the paper is rejected.

**Reviewer Concerns:**

The community encourages proposing new foundational tasks and challenges, but the primary distinction between VC and other tasks lies in the differing scenarios. I believe this is insufficient.

**Reviewer Scores:**

Even with full discussion, the necessity of the task and the evaluation metrics remain below the acceptable threshold.

---

### Decision · Program_Chairs · 2026-01-26

Reject